# Occurrence of Multiple *stx1* Genes and Rare Genomic Variation in *stx1* Shiga Toxin-Producing *Escherichia coli*

**DOI:** 10.3390/microorganisms13051079

**Published:** 2025-05-06

**Authors:** Michaela Projahn, Maria Borowiak, Matthias Contzen, Ekkehard Hiller, Christiane Werckenthin, Elisabeth Schuh, Carlus Deneke

**Affiliations:** 1National Reference Laboratory for *Escherichia coli* Including VTEC, German Federal Institute for Risk Assessment, 12277 Berlin, Germany; 2National Study Centre for Sequencing in Risk Assessment, German Federal Institute for Risk Assessment, 12277 Berlin, Germany; 3Chemical and Veterinary Analysis Agency Stuttgart, 70736 Fellbach, Germany; 4Lower Saxony State Office for Consumer Protection and Food Safety (LAVES), Food and Veterinary Institute, 26133 Oldenburg, Germany

**Keywords:** *Escherichia coli*, STEC, stx, genome

## Abstract

Shiga toxin-producing *Escherichia coli* are important foodborne pathogens. There are several subtypes of the Shiga toxin Stx known, with Stx2 (a–o) being more diverse than Stx1 (a, c, d). Multiple occurrences of *stx2* genes as well as combinations of *stx1* and *stx2* have been reported. However, there is a lack of knowledge on the occurrence of multiple *stx1* genes in STEC strains. Here, we report two strains from food and animal feces which show genomic variations in the *stx1* operon. The first strain harbors *stx1a* and *stx1c* genes, and the second strain shows an inactive *stx1* operon due to an insertion in the *stxA1a* subunit gene. The screening of publicly available complete genome sequences of STEC revealed further strains harboring multiple *stx1* genes, indicating that those strains also occur in human infections. This should be kept in mind when applying routine diagnostic methods like PCR, that do not detect multiple occurrences of *stx1* genes of the same subtype. Moreover, the impact on the severity of human infections due to multiple *stx1* genes has not been investigated well.

## 1. Introduction

Shiga toxin-producing *Escherichia coli* are important foodborne pathogens worldwide. They are naturally occurring in the gastrointestinal tracts of cattle and small and wild ruminants, as well as other mammals, fish, birds and some insects [1], and can cause diarrhea and hemolytic uremic syndrome (HUS) in humans [2]. The main virulence factor—the Stx toxin—is an AB5 toxin which is encoded by the *stxA* and *stxB* subunit genes [3]. Currently, there are 3 subtypes reported for Stx1 toxins (a, c, d) and 15 subtypes are known for Stx2 toxins (a–o) [4,5,6,7], showing a much higher diversity in *stx2* genes than in *stx1* genes. The occurrence of multiple *stx2* genes has previously been reported in studies on STEC [8]. Also, the combination of *stx1* and *stx2* genes in one strain has been clearly proven [9]. In contrast, the occurrence of multiple *stx1* genes in STEC has not been well investigated and has not been explicitly mentioned in STEC studies to the best of our knowledge. The combination of *stx1* genes is also not included in risk analyses on public health risks based on the occurrence of *stx* subtypes and further virulence genes [9]. When it comes to genomic variation or inactivation in *stx* genes, the inactivation of *stx2* genes via the insertion of transposases into the subunit genes has previously been reported [10,11,12]. However, similar events have not been specifically described for *stx1* genes before.

Here, we analyzed two STEC strains which were isolated from food and animal feces in Germany, showing genomic events in terms of *stx1* gene combination and genomic variation. We also analyzed sequence data from public repositories concerning further occurrences of multiple *stx1* combinations in STEC strains.

## 2. Materials and Methods

The two STEC strains were isolated by Federal state laboratories from food and animal source programs according to DIN CEN ISO/TS13136:2013-04 [13] in the framework of German food control and Zoonosis monitoring. Strain BfR-EC-20006 (CVUAS 31424.2) was isolated from roe deer meat in 2023 using modified Tryptic Soy Broth with Novobiocin (Merck KGaA, Darmstadt, Germany). Strain BfR-EC-18115 (LVI OL 10-12119-00272) was isolated from a feces sample of a c. eight-month-old cow, taken at a slaughterhouse in 2019. Isolation was performed using buffered peptone water. The strains were characterized at the National Reference Laboratory for *Escherichia coli* including VTEC (NRL-*E. coli*) and the National Study Centre for Sequencing in Risk Assessment using short-read and long-read genomic sequencing. For this purpose, whole genomic DNA was isolated using the PureLink^®^ Genomic DNA Mini Kit (Invitrogen, Carlsbad, CA, USA) according to the manufacturers’ instructions.

The libraries for short-read sequencing were prepared using the llumina DNA Prep, (M) Tagmentation kit (Illumina, San Diego, CA, USA). The libraries were sequenced on the Illumina MiSeq benchtop sequencer using the MiSeq Reagent Kit v3 (600 cycles; Illumina) in 2 × 201 bp cycles or the NextSeq500 benchtop sequencer using the NextSeq 500/550 Mid Output Kit v2.5 (300 Cycles) in 2 × 151 bp cycles (Illumina, San Diego, CA, USA). For long-read sequencing, genomic DNA was subjected to an additional purification step using Ampure XP (Beckman Coulter GmbH, Krefeld; Germany). DNA was adjusted to 2000 ng in 100 µL with ultrapure water in a 1.5 mL reaction tube. The beads were added to the samples (0.45 times the sample volume), mixed by flicking, and briefly centrifuged. The samples were incubated for 5 min at room temperature. The tubes were placed on a magnetic stand, and the supernatant was discarded. The bead pellets were washed twice with 80% ethanol. Remaining ethanol was completely removed and the pellet was dried for 30 s. Elution was performed in 16 µL 10 mM Tris-HCl pH 8.0 for 15 min at 37 °C. For long-read sequencing, library preparation was carried out using the Rapid Barcoding Kit 96 V14 (Oxford Nanopore Technologies, Oxford, UK). Sequencing was performed on a MinIon Mk1C device using a R10.4.1 flow cell (Oxford Nanopore Technologies, Oxford, UK). Signal data (fast5) were basecalled using guppy v6.4.8 in SUP mode. Long-read data were assembled using the MiLongA pipeline (https://gitlab.com/bfr_bioinformatics/milonga). There, reads were trimmed and filtered using porechop v0.2.4 (https://github.com/rrwick/Porechop) and nanofilt v2.8.0 [14]. Subsequently, fastq data were subjected to assembly using flye v2.9 [15]. Resulting assemblies were polished with Illumina short-read data using pilon v1.24 [16].

For the investigation of public data concerning multiple occurrences of *stx1* genes, all available complete *Escherichia coli* genomes from NCBI RefSeq (https://www.ncbi.nlm.nih.gov/refseq/, accessed on 6 March 2025) [17] were retrieved. The presence of *stx* genes was determined using abricate (https://github.com/tseemann/abricate, with mincov = 50, minid = 80) against a custom database of full-length stx genes. Likewise, complete NCBI Genbank genomes were also scanned. Genomes were selected that carried two or more (nearly) complete *stx* genes.

Strain characterization was performed using BakCharak (https://gitlab.com/bfr_bioinformatics/bakcharak), which implements the determination of the genoserotype based on the CGE SeroTypeFinder [18] and the EcOH database [19] as well as the MLST typing tool mlst (https://github.com/tseemann/mlst), the virulence finder database [20] and AMRFinderPlus v4.0 [21]. Sequence annotation was performed using bakta [22]. Sequence comparisons of *stx* genes with respective reference genes were carried out with Geneious Prime^®^ (version 2024.0.7., Biomatters Ltd., Auckland, New Zealand). The flanking inverted repeats of the transposon were identified using ISFinder (https://isfinder.biotoul.fr/) and the *stx1* phage integration sites were determined using PHASTEST (https://phastest.ca).

## 3. Results

### 3.1. STEC Strains from Food and Livestock in Germany

STEC strains were isolated from food and food-producing animals in the framework of German food control and the Zoonosis monitoring programs (Table 1).

Strain BfR-EC-20006 (CVUAS 31424.2) was isolated from the meat of deer and was assigned as genoserotype O128:H2. For this strain, two different *stx1* subtype genes, namely *stx1a* and *stx1c*, were detected. Strain BfR-EC-18115 (LVI OL 10-12119-00272) was isolated from cattle feces and was determined as O150:H2. This strain harbors an inactive form of the *stx1a* operon due to an insertion element in the *stxA1a* subunit (Figure 1) but also an additional *stx2a* gene.

### 3.2. Sequence Data from Public Repositories

We found indications that *stx1a* and *stx1c* can occur in duplicate or even triplicate (Table 2). Moreover, *stx1a* frequently occurs together with *stx2a* or *stx2c* (see Appendix A). However, only seven genomes could be identified harboring multiple *stx1* genes. All of the seven publicly available strains were reported to be isolated from humans between 2008 and 2023 in different world regions. They were associated with different serotypes and MLST and frequently harbored additional virulence factors such as *eae, nleB* and *ehxA*.

Strains 180-PT54 and 644-PT8 were sequenced in the frame of an outbreak investigation and were characterized concerning short-term evolution but were not investigated concerning the duplication of *stx1* genes [23]. Strains 2013C-3033 and 2013C-4974 were published within a study on PacBio sequencing of STEC [24]. Strain PV0838 was investigated within a study on STEC O26:H11 strains and the authors mentioned the duplication of the *stx1a* genes and also investigated the respective integrations sites of both phages [25]. Strain PNUSAE145590 (BioSample: SAMN36828866) was isolated during an outbreak which was associated with municipal irrigation water, but the investigators only reported on the serotype O157:H7 and not on the *stx* genes [26]. All of the public strain sequences harbored multiple genes of the same *stx1* subtype. Therefore, strains were also checked for variations within these genes (Table 3). It turned out that strains 180-PT54 and 644-PT8 had different variants of the same *stx1* subtype, whereas 2013C-3033, 2013C-4974, PV0838 and PNUSAE145590 harbored identical alleles multiple times in the genomes. In all cases, the locations of the different *stx* genes were separated by more than 100 kbp. In the case of PNUSAE145590, the second *stx1a* gene was localized on another contig, which was most likely circularly assembled phage DNA instead of a plasmid.

## 4. Discussion

STECs are important food-borne pathogens showing high diversity in terms of *stx* gene content in their genomes. In recent years, several new subtypes have been identified for Stx2 toxins [4,5,6,7], whereas for Stx1, only three have been discovered in *E. coli* so far. This indicates a much higher frequency of genomic rearrangement including certain recombination and mutation events in *stx2* genes than in *stx1* genes.

Risk analysis studies revealed that infections with STEC harboring certain subtypes like *stx2a, stx2d* and *stx2c*, especially in combination with the virulence gene *eae*, are more likely to lead to severe symptoms [9]. Therefore, more research is being conducted and efforts are being invested in studying *stx2* variants to gain a better understanding and improve diagnostics. It turned out that other STECs can also be a threat to human health [7,27]. STECs in general can cause a wide variation in symptoms in humans, ranging from no or only mild symptoms to HUS, kidney failure and also death [2].

Routine diagnosis using PCR methods is able to detect a certain *stx* gene or *stx* subtype. However, it is not possible to provide information on the number of genes of a particular *stx* subtype in a single strain. With the increasing use of next generation sequencing (NGS) in bacterial analyses, STEC strains can also be characterized in more detail, including the detection of multiple occurrences of genes of the same *stx* subtype or the integration of insertion sequences/transposases into *stx* subunit genes, which are seldom-reported genomic events but might also be underestimated [10,11,12]. For the investigation of gene duplications and insertion sequence/transposon integration, short-read data provide only limited insights, since such events often lead to contig breaks in the assemblies and therefore cannot be resolved. Hence, long-read sequencing must be performed for the detection of such genomic events. Therefore, we screened only complete genomes from public repositories. However, although NGS methods and bioinformatics tools have improved over the years, specific results cannot be reliably assessed without a comprehensive examination of the methods and tools used for each individual sequence.

Here, we report that *stx1* genes can also occur multiple times in STEC genomes, and it seems like the subtypes *stx1a* and *stx1c* are likely involved in multiplications, but not the subtype *stx1d*. This topic is yet not well investigated, and raises the question of whether multiple occurrences can have an influence on the severity of human infections. Moreover, we showed that the integration of insertion sequences occurs not only in *stx2* subunit genes but also in *stx1* subunit genes, giving insights into genomic rearrangements in STECs harboring *stx1* genes.

## Figures and Tables

**Figure 1 microorganisms-13-01079-f001:**
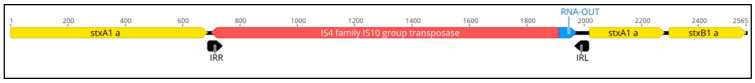
Genomic region of the *stx1a* operon of strain BfR-EC-18115 showing the integration of an insertion element into the *stxA1a* subunit gene (yellow: *stx1a* subunit genes, red: transposase, blue: non-coding RNA, black: inverted repeats, numbers represent base pairs [bp]).

**Table 1 microorganisms-13-01079-t001:** STEC strains from food.

Strain	Laboratory Number	*stx* Genes	*stx1*-Phage Integration Sites	Serotype	MLST	Additional VF	Isolation Source	Isolation Year
BfR-EC-20006	CVUAS 31424.2	*stx1a + stx1c*	*stx1a*: *trnR*, *stx1c*: *dmsB*	O128:H2	ST306	*eae*, *nleB*, *ehxA*	Meat, roe deer	2023
BfR-EC-18115	LVI OL 10-12119-00272	*stx1a* with insertion + *stx2a*	*yehR*	O150:H2	ST25	*ehxA*	Feces, cattle	2019

**Table 2 microorganisms-13-01079-t002:** Overview of STEC strain characteristics derived from public NCBI data.

Accession No.	Strain	*stx* Genes	*stx1*-Phage Integration Sites	Serotype	MLST	Additional VF	Isolation Source	Isolation Year	Submitting Country
GCF_001650275.1	180-PT54	2x *stx1a* + *stx2c*	*cscA-cscK*, *mrlA-fimB*	O157:H7	ST11	*eae*, *nleB*	Stool, human diarrhea	2012	UK
GCF_001650295.1	644-PT8	2x *stx1a* + *stx2c*	*cscA-cscK*, *mrlA-fimB*	O157:H7	- ^#^	*eae*, *nleB*	Stool, human diarrhea	2012	UK
GCF_003017765.1	2013C-3033	2x *stx1c*	*dmsB*, *eco-mqo*	O146:H21	ST442	*none*	Stool, human	NA	USA
GCF_003018375.1	2013C-4974	3x *stx1a*	*ybhC-int*, *yccE-agp*, *phoQ*	O5:H9	ST342	*eae*, *nleB*, *ehxA*	Stool, human	NA	USA
GCF_027925805.1	PV0838	2x *stx1a + stx2d*	*torA, host specificity protein J*	O26:H11	ST21	*eae*, *ehxA*, *nleB*	Human diarrhea	2008	Japan
GCF_030908665.1	PNUSAE145590	2x *stx1a +* 2x *stx2a*	*mlrA*, ***	O157:H7	ST11	*eae*, *ehxA*, *nleB*	human	2023	USA

MLST—Multi-locus sequence type, NA—not available. VF—Virulence factors, ^#^ not yet defined, * possibly circularized phage DNA.

**Table 3 microorganisms-13-01079-t003:** Comparison of *stx1* allele sequences of public data (*stx1* duplicated variants only).

Accession No	Strain	*stx1* Gene	Coverage in %	Identity in %	Genomic Location in bp	Reference Sequence
GCF_001650275.1	180-PT54	*stx1a*	100.00	99.84	1580531–1581757	Z36900
GCF_001650275.1	180-PT54	*stx1a*	100.00	99.92	1881662–1882888	AF034975
GCF_001650295.1	644-PT8	*stx1a*	100.00	99.84	1579718–1580944	Z36900
GCF_001650295.1	644-PT8	*stx1a*	100.00	99.92	1880849–1882075	AF034975
GCF_003017765.1	2013C-3033	*stx1c*	99.92	99.92	1480119–1481345	AJ312232
GCF_003017765.1	2013C-3033	*stx1c*	100.00	100.00	2356992–2355765	AJ312232
GCF_003018375.1	2013C-4974	*stx1a*	100.00	100.00	540125–541351	AF034975
GCF_003018375.1	2013C-4974	*stx1a*	100.00	100.00	1077844–1079070	AF034975
GCF_003018375.1	2013C-4974	*stx1a*	100.00	100.00	2086555–2085329	AF034975
GCF_027925805.1	PV0838	*stx1a*	100.00	99.92	1618814–1620040	AF034975
GCF_027925805.1	PV0838	*stx1a*	100.00	99.92	2720629–2721855	AF034975
GCF_030908665.1	PNUSAE145590	*stx1a*	100.00	99.92	1: 1745155–1746381	Z36900
GCF_030908665.1	PNUSAE145590	*stx1a*	100.00	99.92	2: 20865–22091 *	Z36900

* Possibly circularized phage DNA.

## Data Availability

Sequence data are available under BioProject PRJNA1234488 (BioSample number SAMN47290620 and SAMN47290621).

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
