# Peer review of "Occurrence of Multiple *stx1* Genes and Rare Genomic Variation in *stx1* Shiga Toxin-Producing *Escherichia coli"

_microorganisms, 2025, doi:10.3390/microorganisms13051079_

Round 1

Reviewer 1 Report

Comments and Suggestions for Authors

The authors of “Occurrence of multiple stx1 genes and rare genomic variation in stx1 Shiga toxin-producing Escherichia coli” have written an interesting manuscript about the occurrence of multiple stx1 genes and inactivation of either of their subunits, via insertion of transposases, in STEC isolates. These are already well known phenomena for stx2 genes, but info for stx1 is missing.
The manuscript is easy to read and follow, but I still have some comments and suggestions.

Unfortunately, the Materials and Methods section misses a description of the isolation of the two German isolates investigated; from which source, what year, which protocol was followed to obtain them. 
Please correct the mixed use of past and present tense in single sentences, for example in lines 119-121 and lines 136-139.
Why is the BioSample number mentioned for PNUSAE145590 and not for the others also extracted from the public repository?
I find the sentence in lines 149-151 somewhat odd after the description of the subtype variants in lines 145-149. It fits better with the ones in the next section Please adjust. Moreover, I would suggest to start the discussion about your own work, so what is written in paragraph 3 of this section.

Minor comments
Line 55; states the use of long-read genomic sequencing for the characterization of the two strains, while in the section in lines 58-78 also Illumina sequencing is mentioned. Please correct.
Lines 85-87; Please remove “and subsequently subjected to BakCharak analysis (https://gitlab.com/bfr_bioinformatics/bakcharak)”, because this is also stated in the next lines 88-89.
Line 118, Please remove the word isolated after humans,
In Table 2 some Additional VFs are not written in italic.
Line 137; Please remove and after 644-PT8
Line 147; whereas for Stx1 only three have been
Lines 180-184; Please write the stx genes in italic even if it concerns the Supplementary Materials

Author Response

Reviewer responses

We’d like to thank the reviewers for taking the time to review this manuscript. Please find the detailed responses below and the corresponding revisions highlighted in the re-submitted files.

Reviewer 1

Unfortunately, the Materials and Methods section misses a description of the isolation of the two German isolates investigated; from which source, what year, which protocol was followed to obtain them.

Response: Thank you for pointing this out. We agree with this comment and added the respective isolation protocol in lines 53-58. Isolation source and year of the isolates are mentioned in table 1.

Please correct the mixed use of past and present tense in single sentences, for example in lines 119-121 and lines 136-139.

Response: done

Why is the BioSample number mentioned for PNUSAE145590 and not for the others also extracted from the public repository?

Response: We added the BioSample number for this strain for easily identification in the respective reference as Osborn et al., used the BioSample numbers instead of the strain names/identifiers in their manuscript. All the other strains can be identified by their strain name also used in NCBI.

I find the sentence in lines 149-151 somewhat odd after the description of the subtype variants in lines 145-149. It fits better with the ones in the next section Please adjust.

Response: Agree. We have, accordingly, modified it and moved the sentence to the next section.

Moreover, I would suggest to start the discussion about your own work, so what is written in paragraph 3 of this section.

Response: We thank the reviewer for the suggestion, but the authors would like to maintain the current structure of the discussion.

Minor comments
Line 55; states the use of long-read genomic sequencing for the characterization of the two strains, while in the section in lines 58-78 also Illumina sequencing is mentioned. Please correct.

Response: done. Short read sequencing was added to the sentence in line 55.

Lines 85-87; Please remove “and subsequently subjected to BakCharak analysis (https://gitlab.com/bfr_bioinformatics/bakcharak)”, because this is also stated in the next lines 88-89.

Response: done

Line 118, Please remove the word isolated after humans,

Response: done

In Table 2 some Additional VFs are not written in italic.

Response: done

Line 137; Please remove and after 644-PT8

Response: done

Line 147; whereas for Stx1 only three have been

Response: done

Lines 180-184; Please write the stx genes in italic even if it concerns the Supplementary Materials

Response: done

Reviewer 2 Report

Comments and Suggestions for Authors

Nice small report about strains with multiple stx1 genes.

Minor comments:

Line 31-32: the authors suggest that STEC are only carried by ruminants; this is not true. See the first sentence of thee abstract of their reference 1. Please modify this sentence.

Supplementary materials: it should be easier for the reader to repeat the legend in the file.

Author Response

Reviewer responses

We’d like to thank the reviewers for taking the time to review this manuscript. Please find the detailed responses below and the corresponding revisions highlighted in the re-submitted files.

Reviewer 2:

Line 31-32: the authors suggest that STEC are only carried by ruminants; this is not true. See the first sentence of thee abstract of their reference 1. Please modify this sentence.

Response: Agree. We have, accordingly, modified the sentence.

Supplementary materials: it should be easier for the reader to repeat the legend in the file.

Response: Thank you for pointing this out. We agree with this comment and added the legend to the supplementary file.

Round 2

Reviewer 1 Report

Comments and Suggestions for Authors

The authors have adequately responded to my suggestions/comments in my review of their original manuscript.

Minor comment
In line 143 there appears to be double spaces in between likely and circularly

Reviewer 2 Report

Comments and Suggestions for Authors

The authors answered adequately to my remarks.